# A Closer Look at Dual Batch Normalization and Two-domain Hypothesis In Adversarial Training With Hybrid Samples

## Abstract

There is a growing concern about applying batch normalization (BN) in adversarial training (AT), especially when the model is trained on both *adversarial* samples and *clean* samples (termed Hybrid-AT). With the assumption that *adversarial* and *clean* samples are from two different domains, a common practice in prior works is to adopt dual BN, where $BN_{adv}$ and $BN_{clean}$ are used for adversarial and clean branches, respectively. A popular belief for motivating dual BN is that estimating normalization statistics of this mixture distribution is challenging and thus disentangling it for normalization achieves stronger robustness. In contrast to this belief, we reveal that what makes dual BN effective mainly lies in its two sets of affine parameters. Moreover, we demonstrate that the domain gap between adversarial and clean samples is actually not very large, which is counter-intuitive considering the significant influence of adversarial perturbation on the model. Overall, our work sheds new light on understanding the mechanism of dual BN in Hybrid-AT as well as its underlying two-domain hypothesis. Recommended practices are summarized as takeaway insights for future practitioners.

## 1 Introduction

Adversarial training (AT) (Madry et al., 2018) that optimizes the model on adversarial examples is a time-tested and effective technique for improving robustness against adversarial attack. Beyond classical AT (also termed Madry-AT) (Madry et al., 2018), a common AT setup is to train the model on both *adversarial* samples and *clean* samples (termed Hybrid-AT) (Goodfellow et al., 2015; Kannan et al., 2018; Xie et al., 2020a). Batch normalization (BN) (Ioffe & Szegedy, 2015) has become a de facto standard component in modern deep neural networks (DNNs) (He et al., 2016; Huang et al., 2017; Zhang et al., 2019a; 2021), however, there is a notable concern regarding how to use BN in the Hybrid-AT setup. The concern mainly stems from a two-domain hypothesis: "clean images and adversarial images are drawn from two different domains" (Xie & Yuille, 2020). Guided by this hypothesis, a technique has been proposed to disentangle the mixture distribution of the two domains by applying a separate BN for each domain (Xie & Yuille, 2020).

The above technique has been adopted in multiple works with different names, such as auxiliary BN (Xie et al., 2020a), mixture BN (Xie & Yuille, 2020), Dual BN (Jiang et al., 2020; Wang et al., 2020; 2021). Despite different names, they refer to the same practice of adopting $BN_{adv}$ and $BN_{clean}$ for adversarial and clean samples, respectively. To avoid confusion, we stick to using Dual BN for the remainder of this work. Despite its increasing popularity, the mechanism of how dual BN helps Hybrid-AT remains not fully clear. Towards a better understanding of the underlying mechanism, we first revisit a long-held belief motivated by the two-domain hypothesis (Xie & Yuille, 2020). Specifically, (Xie & Yuille, 2020) justifies the necessity of dual BN in hybrid AT with the following claim (quoted from the abstract of (Xie & Yuille, 2020)):

*"Estimating normalization statistics of the mixture distribution is challenging"* and *"disentangling the mixture distribution for normalization, i.e., applying separate BNs to clean and adversarial images for statistics estimation, achieves much stronger robustness."*

The underlying motivation for the above claim is that BN statistics calculated on clean domain are incompatible with training the model on adversarial domain, and vice versa. Therefore, Hybrid-AT

with single BN suffers from such incompatibility with the mixed distribution for calculating the normalization statistics. Meanwhile, it is claimed in (Xie & Yuille, 2020) that this incompatibility can be avoided by Dual BN through training the clean branch on $\text{BN}_{clean}$ and the adversarial branch on $\text{BN}_{adv}$. As a preliminary investigation of this incompatibility, our work experiments with a new variant of AT with cross-domain BN, namely training the adversarial branch with $\text{BN}_{clean}$. Interestingly, we find that using BN from another domain only has limited influence on the performance. This observation inspires us to have closer look at what component in dual BN makes it more effective than single BN in Hybrid-AT. Through untwining normalization statistics (NS) and affine parameters (AP) in dual BN to include one effect while excluding the other, we demonstrate that disentangled AP instead of NS plays the main role in the merit of dual BN in Hybrid-AT. Moreover, we find that different APs in dual BN also well explain the performance discrepancy caused by the BN choice (either $\text{BN}_{adv}$ or $\text{BN}_{clean}$) at test time, which refutes prior claim which mainly attributes it to the role of NS (Xie & Yuille, 2020).

The motivation for introducing Dual BN is inspired by a two-domain hypothesis that "clean images and adversarial images are drawn from two different domains" (Xie & Yuille, 2020). After showing their motivation does not hold, we further revisit this two-domain hypothesis itself. We reveal that the domain gap between adversarial and clean samples is not as large as claimed in prior work (Xie & Yuille, 2020). We point out a hidden flaw when prior work visualizes the NS from two domains for highlighting a large adversarial-clean domain gap. Specifically, their visualization ignores the influence of different AP. After fixing this hidden flaw, we demonstrate that this domain gap is small. Interestingly, under the same perturbation/noise magnitude, we show that there is no significant difference between adversarial-clean domain gap and noisy-clean counterpart. Therefore, we propose a two-task hypothesis to replace the two-domain hypothesis in (Xie & Yuille, 2020; Xie et al., 2020a) for theoretical justification on the necessity of dual BN in Hybrid AT. We design a dual linear classifier experiment to verify this two-domain hypothesis which also motivates us to apply dual AP to architectures with other normalization modules. Beyond vanilla Hybrid-AT, we further experiment with Trades-AT (another variant of Hybrid-AT) (Zhang et al., 2019b) which by default adopts single BN. We point out an NS inconsistency issue in their original implementation and demonstrate that fixing it can significantly improve performance. Moreover, we find that the KL regularization loss in Trades-AT can also be introduced to improve vanilla Hybrid-AT in the single BN setting.

The model robustness under PGD-10 attack (PGD attack with 10 steps) and AutoAttack (AA) (Croce & Hein, 2020) are evaluated in our analysis as the basic experimental settings, with more details reported in Section A of the appendix and a more specific setup discussed in the context. Overall, considering the increasing interest in adopting dual BN in Hybrid-AT, our work comes timely by taking a closer look at dual BN in Hybrid-AT as well as its underlying hypothesis for theoretical justification. The main findings of our investigation are summarized as follows:

- In contrast to prior work that attributes the merit of dual BN in Hybrid-AT to disentangling NS, we demonstrate that what plays the major role lies in its two sets of AP.

- The claimed large domain gap in prior work is caused by a hidden flaw of ignoring the impact of AP, which motivates a two-task hypothesis for interpreting the task of Hybrid-AT.

- As takeaways, we recommend NOT disentangling NS in Hybrid-AT, since disentangling NS has little influence on performance with dual AP and actually harms performance in the single AP setting. Moreover, with a careful choice of training details, a single BN might be sufficient for achieving competitive performance.

## 2 PROBLEM OVERVIEW AND RELATED WORK

### 2.1 DEVELOPMENT OF ADVERSARIAL TRAINING

**Adversarial training.** Adversarial training (AT) has been the most powerful defense method against adversarial attacks, among which Madry-AT (Madry et al., 2018) is a typical method detailed as follows. Let's assume $\mathcal{D}$ is a data distribution with $(x, y)$ pairs and $f(\cdot, \theta)$ is a model parametrized by $\theta$. $l$ indicates cross-entropy loss in classification. Instead of directly feeding clean samples from $\mathcal{D}$ to minimize the risk of $\mathbb{E}_{(x,y)\sim\mathcal{D}}[l(f(x,\theta),y)]$, (Madry et al., 2018) formulates a saddle problem for

finding model parameter $\theta$ by optimizing the following adversarial risk:

$$\arg \min_{\theta} \mathbb{E}_{(x,y)\sim\mathcal{D}} \left[ \max_{\delta\in\mathbb{S}} l(f(x + \delta;\theta), y) \right] \tag{1}$$

where $\mathbb{S}$ denotes the allowed perturbation budget which is a typically $l_p$ norm-bounded $\epsilon$. We term the above adversarial training framework as Classical-AT. It adopts a two-step training procedures(inner maximization + outer minimization), and trains the robust model with only adversarial samples. Following the same procedure, (Xie & Yuille, 2020) proposes to train the robust model with both clean and adversarial samples, termed as **Hybrid-AT**. The loss of Hybrid-AT is defined as follows:

$$\mathcal{L}_{Hybrid} = \alpha l(f(x;\theta), y) + (1 - \alpha)l(f(x + \delta;\theta), y) \tag{2}$$

where $x$ and $x + \delta$ indicate clean and adversarial samples, respectively. $\alpha$ is a hyper-parameter for balancing the clean and adversarial branches, is set to 0.5 in this work following (Goodfellow et al., 2015; Xie & Yuille, 2020).

**Development of AT.** Since the advent of Mardy-AT and Hybrid-AT, numerous works have attempted to improve AT from various perspectives. From the data perspective, (Uesato et al., 2019; Carmon et al., 2019; Zhang et al., 2019c) have independently shown that unlabeled data can be used to improve the robustness. From the model perspective, AT often benefits from the increased model capacity of models (Uesato et al., 2019; Xie & Yuille, 2020). (Xie et al., 2020b; Pang et al., 2020; Gowal et al., 2020) have investigated the influence and suggested that a smooth activation function, like parametric softplus, is often but not always (Gowal et al., 2020) helpful for AT. It has been shown in (Pang et al., 2020) that the basic training settings in AT can have a significant influence on the model performance, and suggested a set of parameters for fair comparison of AT methods. If not specified, we follow their suggested parameter settings.

## 2.2 BATCH NORMALIZATION IN AT

**Batch normalization (BN).** We briefly summarize how BN works in modern networks. For a certain layer in the DNN, we denote the feature layers of a mini-batch in the DNN as $\mathcal{B} = \{x^1, ..., x^m\}$. The feature layers are normalized by mean $\mu$ and standard deviation $\sigma$ as:

$$\hat{x}^i = \frac{x^i - \mu}{\sigma} \cdot \gamma + \beta \tag{3}$$

where $\gamma$ and $\beta$ indicates the weight and bias in BN, respectively. To be clear, we refer $\mu$ and $\sigma$ as normalization statistics (NS), $\gamma$ and $\beta$ as affine parameters (AP). During training, NS is calculated on the current mini-batch statistics for the update of model weights. Meanwhile, a running average of NS is recorded in the whole training process, which is applied for inference after training ends.

**Dual BN in AT.** There is an increasing interest in investigating BN in the context of adversarial robustness (Awais et al., 2020; Cheng et al., 2020; Nandy et al., 2021; Sitawarin et al.; Gong et al., 2022). This work focuses on Hybrid-AT with dual BN (Xie & Yuille, 2020; Xie et al., 2020a) which applies $\text{BN}_{clean}$ and $\text{BN}_{adv}$ to clean branch and adversarial branch, respectively. Prior work (Xie et al., 2020a) shows that AEs can be used to improve recognition (accuracy) by adversarial training where AEs are normalized by an independent $\text{BN}_{adv}$. Moreover, (Xie & Yuille, 2020) has shown that adding clean images in adversarial training (AT) can significantly decrease robustness performance, where such negative effects can be alleviated to a large extent by simply normalizing CEs with an independent $\text{BN}_{clean}$. Inspired by their finding, (Jiang et al., 2020) also adopts Dual BN in adversarial contrastive learning, showing that single BN performs significantly worse than Dual BN. Beyond Dual BN, triple BN has been attempted in (Fan et al., 2021) for incorporating another adversarial branch. Recently, (Wang et al., 2021) has also combined Dual BN with Instance Normalization to form Dual batch-and-Instance Normalization for improving robustness. A drawback of applying dual BN in Hybrid-AT lies in the unknown source of samples during inference, which makes it difficult to choose the test BN. Prior work (Xie & Yuille, 2020) interprets the necessity of dual BN from the perspective of an inherent large adversarial-clean domain gap, which implicitly suggests disentangling NS (via dual BN) might be the only solution. Our work revisits how dual BN works in Hybrid-AT and finally proposes a new interpretation from a new two-task perspective, which encourages new directions of overcoming the two-task conflict in Hybrid-AT with appropriate regularization instead of dual BN.

## 3 EXISTING MOTIVATION AND PRELIMINARY INVESTIGATION

**Existing motivation for dual BN.** It is hypothesized in (Xie & Yuille, 2020) that clean and adversarial samples are from two domains, thus Dual BN should be introduced to disentangle the mixed distributions in Hybrid-AT. The basic assumption in (Xie & Yuille, 2020) is that there is a clean-adversarial domain gap so that a separate BN is required for each branch. Compared with single BN to handle the mixed distribution, dual BN allows the two branches to be normalized by their corresponding BN only. In other words, the underlying motivation is that the BN statistics from a different domain has a negative influence on the training which can be avoided with Dual BN by only using statistics calculated on its own domain, *i.e.* $BN_{clean}$ for the clean branch and $BN_{adv}$ for the adversarial branch. It is reported in (Jiang et al., 2020) that the model robustness with $BN_{clean}$ adopted at test time is close to zero, which is also confirmed in our work (see Table 1).

Can $BN_{clean}$ be compatible with the adversarial branch to yield a robust model during inference?

**Cross-AT.** We are interested in the answer to the above question because it has significant implications on whether domain-specific BN statistics is better than the mixed one for Hybrid-AT. There is a possibility that this incompatibility is caused by the fact that the adversarial branch of Hybrid-AT is trained with $BN_{adv}$ instead of $BN_{clean}$. To test the above possibility, we perform a *cross-AT* by replacing classical BN with the module in Figure 1 where the adversarial

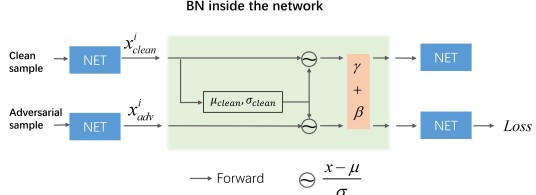

Figure 1: Cross-AT: Replacing $BN_{adv}$ with $BN_{clean}$ in the adversarial branch of Hyrbid-AT.

branch is normalized by $BN_{clean}$. Specifically, the cross-AT still keeps its clean branch in the Hybrid-AT for the forward propagation to get $BN_{clean}$ but the model weights are updated only by the adversarial branch (as shown in Figure 1). Such a setup guarantees that $BN_{clean}$ is used and only used for the adversarial branch. An interpretable meaning of swapping the BN statistics from adversarial to clean is as follows: it constitutes using full cross-domain BN statistics. If the merit of dual BN over single BN lies in replacing mixed BN statistics with domain-specific one, *i.e.* totally avoiding cross-domain statistics, Cross-AT with full cross-domain BN statistics might be expected to yield very low robustness. Interestingly, the results in Figure 2 show that Cross-AT achieves comparable performance as Hybrid-AT with $BN_{adv}$ adopted during inference.

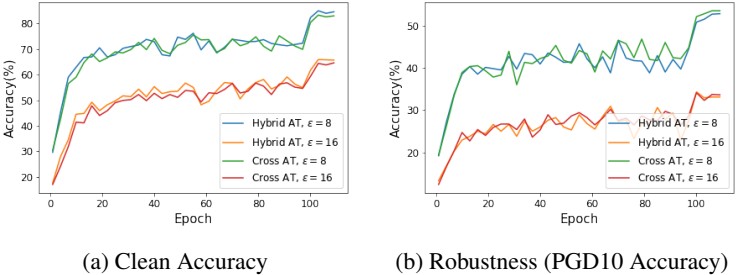

(a) Clean Accuracy         (b) Robustness (PGD10 Accuracy)

Figure 2: Clean and robust accuracy of Cross-AT during training. Hybird-AT is trained with dual BN, while the results of $BN_{adv}$ are reported for comparison since we are mainly interested in whether $BN_{clean}$ can be compatible with the adversarial branch to achieve robustness.

**Remark on the preliminary investigation results.** Such a compatibility of $BN_{clean}$ with the adversarial branch in the above preliminary Cross-AT investigation inspires us to suspect that the merit of dual BN over single BN in Hybrid-AT might not lie in disentangling the mixed distribution for normalization so that each branch is normalized by the statistics calculated on its own domain.

## 4 A CLOSER LOOKER AT DUAL BN IN HYBRID-AT

Introducing an auxiliary BN component, Dual BN causes two changes: (i) disentangling the mixture distribution for normalization statistics (NS) and (ii) introducing two sets of affine parameters (AP).

Prior works (Xie & Yuille, 2020; Xie et al., 2020a) mainly highlight the effect of disentangled NS but pay little attention to that of two sets of AP.

**Cross-Hybrid-AT.** Inspired by the compatibility of $BN_{clean}$ with the adversarial branch in Cross-AT, we perform a Hybrid-AT with cross BN statistics. Different from the setup in Section. 3 that only uses a single BN, dual BN has two sets of APs. Therefore, we let each branch use its own AP but train the two branches with NS from another branch. In other words, the clean branch is trained (and tested) with $NS_{adv}$ and the adversarial branch is trained (and tested) with $NS_{clean}$. We term this setup Cross-Hybrid-AT and its results are reported in Table 1. Table 1 shows that swapping NS has little influence on the performance on the two branches, *e.g.* $NS_{clean}$ in Cross Hybrid-AT achieves similar performance with $NS_{adv}$ in Hybrid-AT. Comparing standard Hybrid-AT and Cross-Hybrid-AT, we highlight that they achieve comparable performance for both adversarial branch and clean branch. The original motivation of disentangling NS is to avoid the influence of NS calculated on partial (half) samples from a different domain, however, we show that NS calculated on full samples from a different domain distribution actually has no significant influence on the performance. Motivated by this observation, in contrast to prior works that attribute the merit of dual BN over single BN in Hybrid-AT to disentangled NS, we establish the following hypothesis:

| Model | NS | AP | Clean | PGD-10 | AA |
|---|---|---|---|---|---|
| Hybrid-AT | $NS_{adv}$ | $AP_{adv}$ | 61.84 | 31.67 | 22.51 |
| | $NS_{clean}$ | $AP_{clean}$ | 94.18 | 0.00 | 0.00 |
| Cross Hybrid-AT | $NS_{clean}$ | $AP_{adv}$ | 59.56 | 31.25 | 22.40 |
| | $NS_{adv}$ | $AP_{clean}$ | 93.86 | 0.00 | 0.00 |

Table 1: Cross Hybrid-AT ($\epsilon = 16/255$).

**Conjecture 1.** We conjecture that what makes Dual BN more effective than single BN in Hybrid-AT is mainly caused by two sets of AP instead of disentangled NS.

### 4.1 UNTWINING NS AND AP IN DUAL BN

**Training setup design.** As discussed above, compared with the default Hybrid-AT baseline, Dual BN brings two effects: disentangled NSs and two sets of APs. To determine the influence of each effect on the model performance, we design two setups of experiments to include only one effect while excluding the other. In Setup1, we only include the effect of two sets of APs, by applying two different sets of APs ($\beta_{adv}/\gamma_{adv}$ and $\beta_{clean}/\gamma_{clean}$) in the adversarial and clean branches while using the default mixture distribution for normalization. In Setup2, we only include the effect of two sets of NSs by only disentangling this mixture distribution with two different sets of NSs while making $BN_{clean}$ and $BN_{adv}$ share the same set of APs. The above setups of BNs are summarized in Figure 3.

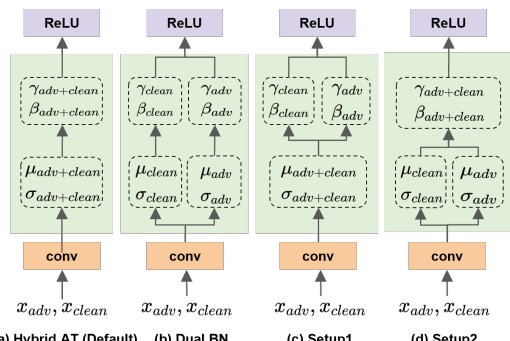

Figure 3: Illustration of different BN setups.

| Setups | NS | AP | $\epsilon = 8/255$ | | | $\epsilon = 16/255$ | | |
|---|---|---|---|---|---|---|---|---|
| | | | Clean | PGD-10 | AA | Clean | PGD-10 | AA |
| Single BN | 1 | 1 | 88.06 | 49.75 | 7.03 | 93.70 | 29.86 | 0.48 |
| Dual BN ($BN_{adv}$) | 2 | 2 | 82.77 | 51.33 | 46.19 | 61.84 | 31.67 | 23.14 |
| Dual BN ($BN_{clean}$) | 2 | 2 | 94.91 | 0.32 | 0.10 | 94.18 | 0.00 | 0.00 |
| Setup1 ($AP_{adv}$) | 1 | 2 | 81.86 | 50.99 | 44.63 | 60.02 | 30.89 | 23.43 |
| Setup1 ($AP_{clean}$) | 1 | 2 | 94.74 | 0.10 | 0.04 | 94.30 | 0.00 | 0.00 |
| Setup2 ($NS_{adv}$) | 2 | 1 | 85.49 | 49.39 | 42.96 | 55.91 | 21.92 | 10.64 |
| Setup2 ($NS_{clean}$) | 2 | 1 | 89.22 | 49.48 | 42.96 | 86.35 | 1.08 | 0.00 |

Table 2: Test accuracy (%) . For NSs, 1 indicates mixture distribution and 2 indicates disentangled distribution for normalziaiton. For APs, 1 indicates single set and 2 indicates double sets of APs. The subscripts of $AP_{adv}$ and $AP_{clean}$ indicate the input data type used during training.

**Results.** As shown in Table 2, Dual BN (with $BN_{adv}$ during inference) brings significant robustness improvement over the Single BN baseline, which is consistent with findings in (Xie & Yuille, 2020). Interestingly, under the attack of PGD-10, their robustness gap is not significant, however, under AA, the Single BN achieves very low robustness (7.03% and 0.48% for $\epsilon = 8/255$ and $\epsilon = 16/255$, respectively). Moreover, Setup1 ($AP_{adv}$) achieves comparable robustness as that of Dual BN ($BN_{adv}$)

for $\epsilon = 8/255$ and $\epsilon = 16/255$, suggesting two sets of APs alone achieve similar performance as dual BN for yielding higher robustness ($AP_{adv}$) than single BN setting. The results collaborate our conjecture that two sets of APs are a key factor in improving Hybrid-AT. The effect of two sets of NSs is more nuanced: for a small perturbation $\epsilon = 8/255$, disentangling mixture distribution is beneficial for boosting the robustness under strong AA; for a large perturbation $\epsilon = 16/255$ this benefit is less significant. This can be explained by the fact the domain distribution mismatch is much smaller for $\epsilon = 8/255$ than that for $\epsilon = 16/255$. Note that Setup2 ($NS_{adv}$) and Setup2 ($NS_{clean}$) achieve comparable robustness for $\epsilon = 8/255$, while their gap is significant for $\epsilon = 16/255$. Overall, we conclude two sets of APs are sufficient for avoiding the issue of low robustness against AA in Default Hybrid-AT, and achieve comparable robustness as Dual BN. Moreover, we experiment with using two sets of AP in either first half or second half of the model, which achieves slightly inferior robustness performance (see Table 11 in appendix).

## 4.2 DOES BN NORMALIZATION STATISTICS CHARACTERIZE DIFFERENT MODEL PERFORMANCE DURING INFERENCE?

Even though dual BN is adopted during training, only one branch of BN parameters can be adopted during inference, and prior works (Xie & Yuille, 2020; Jiang et al., 2020) have reported a large performance gap between $BN_{clean}$ and $BN_{adv}$ during inference. Regarding this phenomenon, prior work claims that "BN normalization statistics (NS) characterizes different model performance" (Xie & Yuille, 2020). In the following, we will refute this claim and show that APs play a major role.

**AP plays a main role in robustness evaluation.** As discussed above, Dual BN not only introduces two sets of NS but also two sets of affine parameters (AP). To investigate the individual role of NS and AP during inference, we conduct a swap experiment during inference as shown in Table 3. The model with original Dual BN achieves a robustness of 51.33% and 0.10 % with $BN_{adv}$ ($NS_{adv}$ and

| Setups | NS | AP | $\epsilon = 8/255$ | | $\epsilon = 16/255$ | |
|---|---|---|---|---|---|---|
| | | | PGD10 | AA | PGD10 | AA |
| Dual BN | $NS_{adv}$ | $AP_{adv}$ | 51.33 | 46.19 | 31.67 | 22.51 |
| | $NS_{clean}$ | $AP_{clean}$ | 0.32 | 0.10 | 0.00 | 0.00 |
| Swap | $NS_{clean}$ | $AP_{adv}$ | 17.1 | 9.16 | 10.02 | 9.80 |
| | $NS_{adv}$ | $AP_{clean}$ | 0.00 | 0.00 | 0.45 | 0.00 |

Table 3: Swap $NS_{clean}$ and $NS_{adv}$ in Hybrid-AT during inference.

$AP_{adv}$) and $BN_{clean}$ ($NS_{clean}$ and $AP_{clean}$), respectively. If we keep $NS_{clean}$ fixed and swap AP, the robustness increases from 0.32% to 17.1% with the $AP_{clean}$ replaced by $AP_{adv}$ . However, when we keep $AP_{clean}$ fixed, both $NS_{clean}$ and $NS_{adv}$ achieve almost zero robustness. These phenomena are strong evidence demonstrating that the APs play a larger part in robustness evaluation. However, the results seem to suggest that NS is also important. For example, the robustness with the configuration of $NS_{clean}$ and $AP_{adv}$ is still much lower than that of $NS_{adv}$ and $AP_{adv}$, for which our analysis in the following section provides an explanation (See Table 4).

## 5 A CLOSER LOOKER AT THE TWO-DOMAIN HYPOTHESIS

A model trained on a source domain performs poorly on a new target domain when there is a domain shift (Daumé III, 2007; Sun et al., 2017). With BN as the target, it is common in the literature (Li et al., 2017; Benz et al., 2021; Schneider et al., 2020; Xie & Yuille, 2020; Xie et al., 2020a) to indicate the domain gap by the difference of NS between two domains. For example, an early work (Li et al., 2017) has shown that adapting NS from the target domain during inference can improve the performance on a new target domain without retraining the model. This test-time BN adaptation has also been adopted in (Benz et al., 2021; Schneider et al., 2020) for improving the model robustness against common corruptions by perceiving them (random noise for instance) as a new domain. With such an understanding, it is straightforward for prior works (Xie & Yuille, 2020; Xie et al., 2020a; Jiang et al., 2020) to also perceive adversarial domain as a new domain.

### 5.1 A HIDDEN FLAW OF VISUALIZING NS IN PRIOR WORK

To highlight the two-domain gap, prior work (Xie & Yuille, 2020) visualizes the difference of NS in $BN_{adv}$ and $BN_{clean}$ (see Figure 5 of (Xie & Yuille, 2020)). We quote the following sentence from (Xie & Yuille, 2020): "*We observe that clean images and adversarial images induce significantly different running statistics, though these images share the same set of convolutional filters for feature extraction*". With our analysis in Section. 4, we know that the AP in $BN_{clean}$ and $BN_{adv}$ is different. The clean branch and adversarial branch still have different weights, *i.e.* AP, even though the same

set of convolutional filters are shared. In other words, the significant difference between $NS_{clean}$ and $NS_{adv}$ is induced by not only the difference between image inputs (clean images *v.s.* adversarial images) but also different model (AP) weights. To summarize, the NS difference between $BN_{clean}$ and $BN_{adv}$ is characterized by two factors: (a) AP inconsistency and (b) different domain inputs.

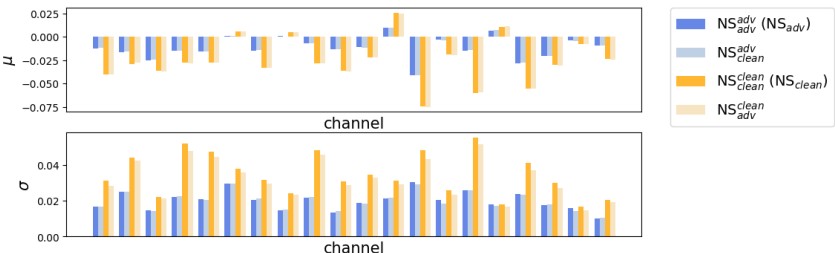

Figure 4: $\mu$ and $\sigma$ visualization. Randomly choose 20 channels and display the NS calculated with different APs.

In the default setup of dual BN, $NS_{clean}$ is calculated on clean samples with $AP_{clean}$, while $NS_{adv}$ is calculated on adversarial samples with $AP_{adv}$. We additionally calculate the NS on clean samples with $AP_{adv}$ (denoted as $NS_{clean}^{adv}$) and calculate the NS on adversarial samples with $AP_{clean}$ (denoted as $NS_{adv}^{clean}$). Following $NS_{adv}^{clean}$ and $NS_{clean}^{adv}$ to indicate AP choice with the superscript and indicate sample choice with the subscript, we can also denote $NS_{clean}$ as $NS_{clean}^{clean}$ and denote $NS_{adv}$ as $NS_{adv}^{adv}$. To exclude the influence of AP inconsistency, we intend to compare NS between clean and adversarial samples with the same AP. In other words, the domain gap is characterized by the difference between $NS_{clean}^{clean}$ and $NS_{clean}^{adv}$ or that between $NS_{adv}^{adv}$ and $NS_{adv}^{clean}$. Following the procedures in (Xie & Yuille, 2020), we plot different types of NS in Figure 4 by randomly sampling 20 channels of the second BN layer in the first residual block. Fig. 4 shows that there exists a gap between $NS_{clean}^{clean}$ and $NS_{adv}^{adv}$, which is consistent with the findings in (Xie & Yuille, 2020). Moreover, there are two other observations from Figure 4. First, if we fix the input samples and calculate NS with different AP, there exists a large gap, *i.e.* the gap between $NS_{clean}^{adv}$ and $NS_{clean}^{clean}$, as well as the gap between $NS_{adv}^{adv}$ and $NS_{adv}^{clean}$. Second, those NSs with the same APs are very close to each other, *e.g.*, $NS_{adv}^{adv}$ and $NS_{clean}^{adv}$ are very similar to each other, and the same applies for $NS_{adv}^{clean}$ and $NS_{clean}^{clean}$. The visualization results highlight the significance of AP in Dual BN, and is consistent with the finding in Section 4. Without considering the influence of AP, the visualization and conclusions in (Xie & Yuille, 2020) might convey a misleading message. We report the visualization results of AP in Figure 7, which shows a significant gap between $AP_{clean}$ and $AP_{adv}$. For a quantitative comparison, we measure the Wasserstein distance between clean and adversarial branches in different layers in the appendix (see Figure 6 in the appendix), which corroborates our above finding.

An interesting phenomenon in Table 3 is that the robustness with $NS_{clean}$ and $AP_{adv}$ achieves much lower robustness(17.1%) than the original $BN_{adv}$ (51.33%). This is caused by the fact that the $NS_{clean}$ is calculated on $AP_{clean}$ and thus there is a mismatch between $NS_{clean}$ and $AP_{adv}$. Following the notations in Fig. 4, we evaluate a pretrained Hybrid-AT model by replacing its original $NS_{clean}$ and $NS_{adv}$ with different NS, as shown in Table 4.

| Setups | NS | AP | PGD10 $\epsilon = 8/255$ | AA | PGD10 $\epsilon = 16/255$ | AA |
|---|---|---|---|---|---|---|
| Default | $NS_{adv}^{adv}$ | $AP_{adv}$ | 51.33 | 46.19 | 31.67 | 22.51 |
| | $NS_{clean}^{clean}$ | $AP_{clean}$ | 0.32 | 0.10 | 0.00 | 0.00 |
| Cross-domain | $NS_{clean}^{adv}$ | $AP_{adv}$ | 51.75 | 46.55 | 32.73 | 24.40 |
| | $NS_{adv}^{clean}$ | $AP_{clean}$ | 0.00 | 0.00 | 0.00 | 0.00 |

Table 4: Cross-domain but **re-calibrated** NS achieves comparable performance.

Table 4 shows that given $AP_{adv}$, $NS_{clean}^{adv}$ achieves a robustness of 51.75%, which is comparable to 51.33% with $NS_{adv}^{adv}$. Moreover, given $AP_{clean}$, both $NS_{adv}^{clean}$ and $NS_{clean}^{clean}$ yield almost zero robustness. We conclude that AP characterizes the large robustness gap between $BN_{clean}$ and $BN_{adv}$ during inference, instead of NS claimed in (Xie & Yuille, 2020). When AP is fixed, the robustness gap between the NS calculated on clean or adversarial samples is quite subtle.

## 5.2 Comparing adversarial-clean and noisy-clean domain gap

As suggested in (Benz et al., 2021; Schneider et al., 2020), noisy samples (images corrupted by random noise) can be seen as a domain different from clean samples. Adversarial perturbation is a *worst-case* noise for attacking the model. Tak-

ing a ResNet18 model trained on clean samples for example, we report the performance under adversarial perturbation and random noise (with the same magnitude) in Table 5. As expected, the model accuracy drops to zero with adversarial perturbation. Under random noise of the same magnitude, we find that the model performance only drops by a small margin. Given that the influence of adversarial perturbation on the model performance is significantly larger than that of random noise, it might be tempting to believe that the adversarial-clean domain gap is much larger than noisy-clean domain gap.

| Noise/perturbation Size | 0 | 8/255 | 16/255 |
|---|---|---|---|
| Random noise | 94.0 | 92.7 | 86.6 |
| Adversarial perturbation | 94.0 | 0.00 | 0.00 |

Table 5: Accuracy (%) under random noise and adversarial perturbation.

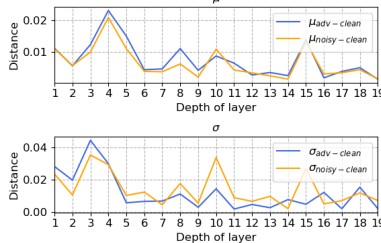

With Wasserstein distance of NS between different domains as the metric, we compare the adversarial-clean domain gap with noisy-clean counterpart on the above ResNet18 model trained on clean samples, as shown in Figure 5. The perturbation and noise magnitude are set to $16/255$ (see Figure 9 for the results of $8/255$). Interestingly, we observe that there is no significant difference between adversarial-clean domain gap and noisy-clean counterpart. In other words, the adversarial-clean domain gap is not as large as many might believe considering the strong performance drop caused by adversarial perturbation.

Figure 5: Comparison of adversarial-clean and noisy-clean domain gap.

### 5.3 INTERPRETING HYRBID-AT FROM A TWO-TASK PERSPECTIVE

**From two-domain to two-task hypothesis.** Considering the adversarial-clean domain gap is similar to noisy-clean domain gap as well as a strict constraint on allowable perturbation budget, future works investigating Hybrid AT are suggested to discard the two-domain hypothesis. Akin to prior work justifying the role of disentangling NSs with the two-domain hypothesis, we provide a two-task hypothesis for justifying the importance of disentangled APs. Intuitively, with the two branches in Hybrid-AT, the model weights are trained for two tasks: one for clean accuracy and the other for robustness. Intuitively, it is difficult for a single same set of parameters to realize two tasks. A common approach for handling two tasks with a shared backbone is to make the top layers unshared. Here, we experiment with a shared encoder of single BN but with dual linear classifiers. The results in Table 6 show that this setup results in similar behavior as dual BN. Such a phenomenon corroborates that disentangling APs is equivalent to making partial learnable network weights not shared between the two tasks.

| Setups | Branch | Clean | PGD10 | AA |
|---|---|---|---|---|
| Dual BN | $BN_{adv}$ | 61.84 | 31.67 | 22.51 |
| | $BN_{clean}$ | 94.18 | 0.00 | 0.00 |
| Dual Linear | $Linear_{adv}$ | 60.72 | 28.84 | 16.50 |
| | $Linear_{clean}$ | 91.43 | 2.21 | 1.30 |

Table 6: Dual linear model, $\epsilon = 16/255$.

**Beyond BN.** In contrast to the two-domain hypothesis, our two-task hypothesis highlights the necessity of disentangling AP instead of NS. This motivates us to apply dual AP into models with other normalization modules when disentangling NS is not applicable. For example, layer normalization (LN) adopts sample-wise NS, and therefore it is not applicable to disentangle distribution-wise NS between two domains. We experiment with dual AP on ResNet and ViT with LN and the results are reported in Table 7 (more results on other normalization are reported in Table 10.) We observe that LN with dual AP results in similar behavior as BN with dual AP, either (b) or (c) in Figure 3, (see Table2). This further corroborates our two-task hypothesis which interprets the main role of dual BN as providing unshared parameters for two tasks. In Section 6, we will show that such a two-task conflict might alternatively be mitigated with an appropriate regularization.

| Model | Norm | Setups | Branch | Clean | PGD10 | AA |
|---|---|---|---|---|---|---|
| ResNet | LN | Single AP | AP | 75.12 | 18.81 | 11.80 |
| | | Dual AP | $AP_{adv}$ | 62.56 | 26.98 | 16.90 |
| | | Dual AP | $AP_{clean}$ | 88.41 | 0.00 | 0.00 |
| ViT | LN | Single AP | AP | 92.21 | 33.60 | 1.84 |
| | | Dual AP | $AP_{adv}$ | 58.02 | 30.08 | 12.44 |
| | | Dual AP | $AP_{clean}$ | 91.60 | 0.00 | 0.00 |

Table 7: Model with Layer Normalization (LN), $\epsilon = 16/255$.

## 6 BEYOND VANILLA HYBRID-AT AND TAKE-AWAY INSIGHT

During inference, whether the test sample is clean or adversarial is unknown and therefore only a single BN can be adopted. Prioritizing the robustness, prior work (Xie & Yuille, 2020) adopts $BN_{adv}$ at test time at the cost of clean accuracy drop. However, with a single BN, it is shown in (Zhang et al.,

2019b) that another variant of Hybrid-AT yields competitive robustness as well as accuracy. Different from the basic loss in Eq 2, the adversarial branch in (Zhang et al., 2019b) is trained by a KL loss and this variant of Hybrid-AT is termed *Trades-AT* for differentiation. Under the perturbation of $\epsilon = 16/255$ ($l_\infty$), unlike vanilla Hybrid AT, the performance of Trade-AT with the default single BN maintains reasonably high robustness but still falls behind its counterpart adopting Dual BN ($\text{BN}_{adv}$ during inference). Interestingly, this robustness gap can be significantly mitigated by fixing a small easily-overlooked training detail, which is pointed out and discussed in the following.

**A closer look at Trades-AT.** With the original implementation of Trades-AT (see Algorithm 1 in Appendix D), clean and adversarial samples are fed to the model independently, which results in $\text{NS}_{adv}$ and $\text{NS}_{clean}$ being disentangled *during training*. During inference, however, the moving-average NS can be seen as a mixture of $\text{NS}_{adv}$ and $\text{NS}_{clean}$. This mixed NS is termed $\text{NS}_{mix}$. This causes a unintended inconsistency among $\text{NS}_{adv}$, $\text{NS}_{clean}$ and $\text{NS}_{mix}$. Even though the gap between $\text{NS}_{adv}$ and $\text{NS}_{clean}$ is not as large as suggested in prior works, their gap still exists especially when the perturbation size is relatively large (16/255). The unintended inconsistency caused by their gap might have a negative effect on the performance. To this end, we experiment with sending clean and adversarial samples together to train the model for removing the NS disentangling effect (see Algorithm 2 in Appendix D), thus $\text{NS}_{mix}$ is used for training both branches as well as test. Somewhat surprisingly, the performance is improved by a large margin after fixing this inconsistency problem (see the performance gap between Single BN (Algorithm 1) and Single BN (Algorithm 2)). For more discussion, see Appendix D. For completeness, we also report the results of Trades-AT with dual BN in Table 8.

| Trades-AT | Clean | PGD10 | AA |
|---|---|---|---|
| Single BN (Algorithm 1) | 69.97 | 26.8 | 18.04 |
| Single BN (Algorithm 2) | 75.14 | 29.41 | 21.32 |
| Dual BN ($\text{BN}_{adv}$) | 62.15 | 33.16 | 25.67 |
| Dual BN ($\text{BN}_{clean}$) | 78.03 | 16.17 | 2.21 |

Table 8: A closer look at Trades-AT with different BN settings, $\epsilon = 16/255$.

After being fixed, Trades-AT (with single BN) adopts the same structure as Hybrid-AT with single BN (see Figure 3 (a)). Their difference lies in whether a KL regularization is adopted to train the adversarial branch. Therefore, we conjecture that a KL regularization in Trades-AT might be a key component for AT on the hybrid samples under the single BN setting. We introduce the KL loss to the Eq 2 and the results in Table 9 show that vanilla Hybrid-AT (Single BN) with KL loss indeed achieves competitive performance for both robustness as well as accuracy to the Dual BN setting.

| Hybrid-AT | Clean | PGD10 | AA |
|---|---|---|---|
| Single BN (Default) | 93.70 | 29.86 | 0.48 |
| Single BN (KL loss) | 68.86 | 33.61 | 23.60 |
| Dual BN ($\text{BN}_{adv}$) | 61.84 | 31.67 | 22.51 |
| Dual BN ($\text{BN}_{clean}$) | 94.18 | 0.00 | 0.00 |

Table 9: Vanilla Hybrid-AT (Single BN) with KL loss, $\epsilon = 16/255$.

**Takeaway insight on disentangling NS and solutions for addressing two-task conflict.** Our work reveals that the gap is not as large as previously visualized (Xie & Yuille, 2020) but it does exist and can cause unintended inconsistency if overlooked by practitioners. As for the recommended practice, the two-domain hypothesis (Xie & Yuille, 2020; Xie et al., 2020a) advocates the practice of disentangling NS. By contrast, we find that disentangling NS has little influence on performance when two sets of AP are used (see Table 2). In the case of a single AP, disentangling NS actually harms the performance. As a takeaway, we recommend the practice of NOT disentangling NS for simplicity. In addition, We show that, with a careful choice of training details, a single BN might be sufficient for achieving competitive performance in the setup of AT with hybrid samples (see Table 8 and Table 9). This suggests promising directions for solving the two-task conflict beyond dual BN.

## 7 CONCLUSION

We experiment with Cross-AT and demonstrate the compatibility of BN statistics of clean samples with the adversarial branch, which inspires to doubt the motivation in prior work for justifying the necessity of dual BN in Hybrid AT. We take a closer look at dual BN and its underlying theoretical hypothesis, which yields two intriguing findings. First, what makes dual BN effective lies in two sets of affine parameters instead of disentangled normalization statistics. Second, the adversarial-clean domain gap is not as large as many might expect and it is similar to its noisy-counterpart under the same perturbation/noise magnitude. In addition, we propose a new interpretation of Hybrid-AT with dual BN from the two task perspective which is shown to generalize to architectures (like ViT) using other variants of normalization modules. Finally, we investigate Hybrid-AT beyond its vanilla version and summarize recommended practices as takeaway insight for future practitioners.

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

## A  EXPERIMENTAL SETUPS

In this work, we perform experiments on CIFAR10 with ResNet18 and follow the suggested training setups in (Pang et al., 2020) unless specified. Specifically, we train the model for 110 epochs. The learning rate is set to 0.1 and decays by a factor of 0.1 at the epoch 100 and 105. We adopt an SGD optimizer with weight decay $5 \times 10^{-4}$. For generating adversarial examples during training, we use $\ell_\infty$ PGD attack with 10 iterations and step size $\alpha = 2/255$. For the perturbation constraint, $\epsilon$ is set to $\ell_\infty$ $8/255$ (Pang et al., 2020) or $16/255$ (Xie & Yuille, 2020). Following (Pang et al., 2020), we evaluate the model robustness under PGD-10 attack (PGD attack with 10 steps) and AutoAttack (AA) (Croce & Hein, 2020).

## B  VISUALIZATION OF NORMALIZATION STATISTICS AND AFFINE PARAMETERS

### B.1  VISUALIZATION OF AFFINE PARAMETERS

As a counterpart of the NS visualization (Figure 4) in Section 5.1 , we visualize $AP_{clean}$ and $AP_{adv}$ in Figure 7, which shows a significant difference between them. For a quantitative comparison, we measure the Wasserstein distance in different layers, as shown in Figure 6. It shows a large distance between $AP_{clean}$ and $AP_{adv}$ in all layers. However, with the same $AP_{adv}$, the gap between $NS_{adv}^{adv}$ and $NS_{clean}^{adv}$ stays almost zero in all layers.

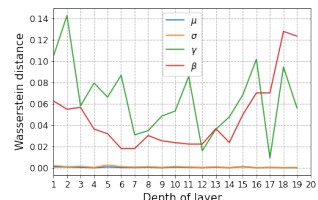

Figure 6: Layer-wise discrepancy visualization.

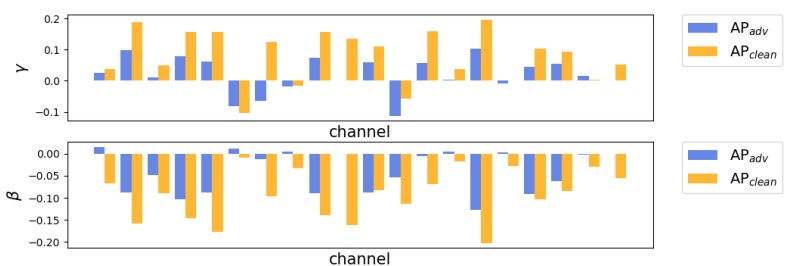

Figure 7: $\gamma$ and $\beta$. Randomly chosen 20 channels for visualizing $\text{AP}_{clean}$ and $\text{AP}_{adv}$.

## B.2 VISUALIZATION OF NS WHEN AFFINE PARAMETERS ARE DISABLED

Figure 8 visualizes the distribution discrepancy between $\text{NS}_{clean}$ and $\text{NS}_{adv}$ when affine parameters are disabled during training and inference. The difference between $\text{NS}_{clean}$ and $\text{NS}_{adv}$ is not as large as visualized in (Xie & Yuille, 2020), which is consistent with the discussions in Section 5.1.

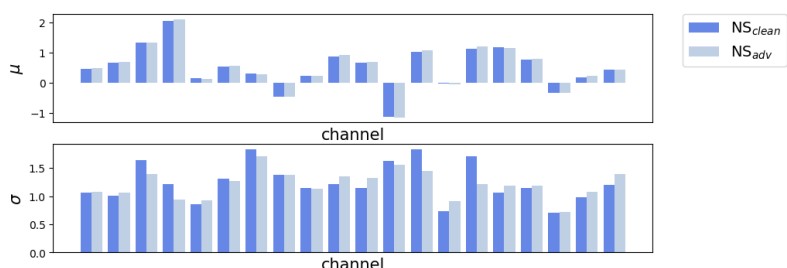

Figure 8: $\mu$ and $\sigma$ when AP is disabled during training and inference. Randomly chosen 20 channels for visualizing $\text{NS}_{clean}$ and $\text{NS}_{adv}$.

## B.3 COMPARISON WITH NOISY-CLEAN DOMAIN GAP

Figure 9 shows the adversarial-clean domain gap with noisy-clean counterpart when perturbation/noise magnitude is $8/255$, which shows the same trend with Figure 5 when perturbation/noise magnitude is $16/255$.

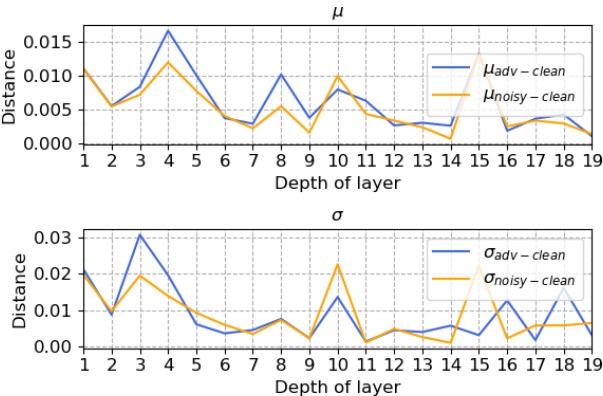

Figure 9: Visualization of adversarial-clean domain gap and noisy-clean domain gap (perturbation/noise magnitude is set to $8/255$.

## C    OTHER RESULTS

### C.1    DUAL AP EXPERIMENTS WITH DIFFERENT NORMALIZATION

In addition to the Layer Normalization (LN) results in Table 7, we report the Dual AP results with Group Normalization (GN) and Instance Normalization (IN) in Table 10. The results in Table 10 show consistent trend with that of BN and LN.

| Model | Norm | Setups | Branch | Clean | PGD10 | AA |
|---|---|---|---|---|---|---|
| ResNet | GN | Single AP | AP | 81.85 | 21.94 | 14.50 |
| | | Dual AP | $AP_{adv}$ | 70.27 | 29.36 | 18.30 |
| | | Dual AP | $AP_{clean}$ | 91.82 | 0.00 | 0.00 |
| | IN | Single AP | AP | 92.55 | 23.06 | 1.20 |
| | | Dual AP | $AP_{adv}$ | 52.29 | 25.27 | 16.10 |
| | | Dual AP | $AP_{clean}$ | 92.35 | 0.00 | 0.00 |

Table 10: Dual AP with Group Normalization (GN) and Instance Normalization (IN), $\epsilon = 16/255$.

### C.2    FURTHER DISENTANGLING AFFINE PARAMETERS

We report the results of using two sets of AP in either first half or second half of the model in Table 11, which achieves slightly inferior robustness performance than Setup1 in Figure 3.

| Setups | AP (first half) | AP (second half) | Clean | PGD-10 | AA |
|---|---|---|---|---|---|
| Setup1 ($AP_{adv}$) | 2 | 2 | 60.02 | 30.89 | 23.43 |
| Setup1 ($AP_{clean}$) | 2 | 2 | 94.30 | 0.00 | 0.00 |
| Setup3 ($AP_{adv}$) | 1 | 2 | 58.14 | 29.45 | 19.80 |
| Setup3 ($AP_{clean}$) | 1 | 2 | 93.66 | 0.00 | 0.00 |
| Setup4 ($AP_{adv}$) | 2 | 1 | 59.91 | 30.62 | 22.10 |
| Setup4 ($AP_{clean}$) | 2 | 1 | 94.05 | 0.00 | 0.00 |

Table 11: Test accuracy (%) with $\epsilon = 16/255$. With 1 set of NS, different sets of AP in the first and second half of the model. The subscripts of $AP_{adv}$ and $AP_{clean}$ indicate the input data type used during training.

### C.3    EVALUATE HYBRID-AT AND CROSS HYBRID-AT WITH DIFFERENT NS DURING INFERENCE

We evaluate Hybrid-AT and Cross Hybrid-AT with different NS during inference, as in Table 12. Table 12 shows that the effect of more data to estimate BN by mixing $NS_{adv}$ and $NS_{clean}$ (noted as $NS_{mix}$) is limited. Under the $AP_{clean}$, the robustness with $NS_{mix}$ stays zero, which is expected. Under the $AP_{adv}$, the robustness with $NS_{mix}$ is slightly lower than the corresponding NS used in training. Note that the gap between $NS_{adv}$ and $NS_{clean}$ is not as large as suggested in prior work but still exists, which explains the performance drop of mixed NS during inference (when disentangled NS is applied during training).

| Model | NS | AP | Clean | PGD-10 | AA |
|---|---|---|---|---|---|
| Hybrid-AT | $NS_{adv}$ | $AP_{adv}$ | 61.84 | 31.67 | 22.51 |
| | $NS_{clean}$ | $AP_{clean}$ | 94.18 | 0.00 | 0.00 |
| | $NS_{mix}$ | $AP_{adv}$ | 61.45 | 29.97 | 18.80 |
| | $NS_{mix}$ | $AP_{clean}$ | 92.00 | 0.00 | 0.00 |
| Cross Hybrid-AT | $NS_{clean}$ | $AP_{adv}$ | 59.56 | 31.25 | 22.40 |
| | $NS_{adv}$ | $AP_{clean}$ | 93.86 | 0.00 | 0.00 |
| | $NS_{mix}$ | $AP_{adv}$ | 60.09 | 26.41 | 18.36 |
| | $NS_{mix}$ | $AP_{clean}$ | 90.60 | 0.00 | 0.00 |

Table 12: Evaluate Hybrid-AT and Cross Hybrid-AT with different NS ($\epsilon = 16/255$).

## D    ON THE IMPLEMENTATION DETAILS OF TRADES-AT

In the original implementation of Trades-AT, As shown in Algorithm 1, the clean samples and adversarial samples are sent to the model independently, which results in $NS_{clean}$ and $NS_{adv}$ being disentangled during training. Moreover, the NS during inference is a moving average of $NS_{clean}$ and $NS_{clean}$, and we term it $NS_{mix}$ for simplicity. As

| Trades-AT | Clean | PGD10 | AA |
|---|---|---|---|
| Default (Algorithm 1) | 69.97 | 26.8 | 18.04 |
| Single BN (Algorithm 2) | 75.14 | 29.41 | 21.32 |
| Single BN (Algorithm 3) | 68.03 | 27.31 | 19.10 |

Table 13: A closer look at Trades-AT with different BN settings, $\epsilon = 16/255$.

pointed out in the main manuscript, the discrepancy among $NS_{adv}$, $NS_{clean}$, and $NS_{mix}$ causes unintended NS consistency. To this end, we experiment with mixing the two types of inputs before feeding them to the model, which is shown in Algorithm 2. To further verify this improvement comes from feeding the clean and adversarial samples **together** to the model, we slightly modify Algorithm 2 to its disentangled version (shown in Algorithm 3). We observe that Algorithm 3 achieves significantly worse performance than that of Algorithm 2. Algorithm 3 performs similarly as Algorithm 1, which is expected because both algorithms feed clean and adversarial samples **independently** to the model.

---

**Algorithm 1** Original implementation of Trades-AT

```
# model: model (e.g., ResNet)
# generate_adv_func: function of generating adversarial samples according to certain
    parameters (e.g., perturbation size)
# cross_entropy_func: function of calculating cross-entropy loss
# kl_func: function of calculating the KL loss.

for (x_clean, label) in loader: # load a minibatch (x_clean, label) with n samples

   x_adv = generate_adv_func(x_clean, model, args)) # generate the adversarial samples

   # calculate cross-entropy loss on clean samples
   logits_clean = model(x_clean) #
   loss_clean = cross_entropy_func(logits_clean, label)

   # calculate KL loss on both clean samples and adversarial samples
   loss_kl = kl_func(log_softmax(model(x_adv)), softmax(model(x_clean)))
   loss = loss_clean + beta * loss_kl

   loss.backward() # back-propagate
   update(model) # update model weight
```

---

**Algorithm 2** Our implementation of Trades-AT by feeding the clean and adversarial samples **together** to the model

```
# model: model (e.g., ResNet)
# generate_adv_func: function of generating adversarial samples according to certain
    parameters (e.g., perturbation size)
# cross_entropy_func: function of calculating cross-entropy loss
# kl_func: function of calculating the KL loss.

for (x_clean, label) in loader: # load a minibatch (x_clean, label) with batch_size
    samples

   x_adv = generate_adv_func(x_clean, model, args)) # generate the adversarial samples

   # mix x_clean and x_adv before calculating their logits (model output)
   x_mix = concat(x_clean, x_adv)
   # feed the clean and adversarial samples together to the model
   logits_mix = model(x_mix)
   # split the logit output for clean branch and adversarial branch
   logits_clean = logits_mix[:batch_size]
   logits_adv = logits_mix[batch_size:]

   # calculate cross-entropy loss on clean samples
   loss_clean = cross_entropy_func(logits_clean, label)

   # calculate KL loss on both clean samples and adversarial samples
   loss_kl = kl_func(log_softmax(logits_adv), softmax(logits_clean))
   loss = loss_clean + beta * loss_kl

   loss.backward() # back-propagate
   update(model) # update model weight
```

**Algorithm 3** Our implementation of Trades-AT by feeding the clean and adversarial samples **independently** to the model

```
# model: model (e.g., ResNet)
# generate_adv_func: function of generating adversarial samples according to certain
    parameters (e.g., perturbation size)
# cross_entropy_func: function of calculating cross-entropy loss
# kl_func: function of calculating the KL loss.

for (x_clean, label) in loader: # load a minibatch (x_clean, label) with batch_size
    samples

    x_adv = generate_adv_func(x_clean, model, args)) # generate the adversarial samples

    # feed the clean and adversarial samples independently to the model
    logits_clean = model(x_clean)
    logits_adv = model(x_adv)

    # calculate cross-entropy loss on clean samples
    loss_clean = cross_entropy_func(logits_clean, label)

    # calculate KL loss on both clean samples and adversarial samples
    loss_kl = kl_func(log_softmax(logits_adv), softmax(logits_clean))
    loss = loss_clean + beta * loss_kl

    loss.backward() # back-propagate
    update(model) # update model weight
```

