# OpenReview forum: "A Closer Look at Dual Batch Normalization and Two-domain Hypothesis In Adversarial Training With Hybrid Samples"
_ICLR.cc/2023/Conference — Submitted to ICLR 2023_

### Official Review · Reviewer_8GPb · 2022-10-24

**Confidence:** 4
**Correctness:** 1
**Technical Novelty And Significance:** 2
**Empirical Novelty And Significance:** 3
**Recommendation:** 5

**Clarity, Quality, Novelty And Reproducibility:**

This paper presents the idea clearly. The hypothesis on affine parameters' impact is novel. The proposed ablation studies should be easy to reproduce.

**Strength And Weaknesses:**

strength:

1. There are many empirical evaluations to support the hypotheses

2. The logic and experiments are easy to follow.

Weakness:

1. This is more like a commentary paper pertaining to an existing paradigm, i.e. Dual BN. Despite some insights into the effectiveness of affine parameters, there is no novel algorithm proposed.

2. From practical point of view, is there any recommended practice that one could adopt based on the analysis? Answering this quesiton could substantially improve the quality of this paper.

3. I am wondering whether this analysis could generalize to modern architectures, e.g. ViT, where BN no longer exists.

Minor:

4. It is recommended to evaluate mixing up NS_adv and NS_clean in Table 1 such that the impact of using more data to estimate BN is eliminated.



**Summary Of The Paper:**

This work reviewed the existing solution of Dual BN for adversarial robustness by focusing on the affine parameters in BN. It is argued that the stronger adversarial robustness in Dual BN is mainly attributed to additional affine parameters. Extensive ablation studies are presented to validate the argument. Moreover, it is argued that domain gap between adversarial and clean samples are not significantly larger than between noisy and clean samples, suggesting new explanations are required.


**Summary Of The Review:**

Overall, this work presents interesting empirical insight into the effectiveness of Dual BN. However, the major weakness lies in the lack of novel solutions. It is hard to grasp useful practical guidelines for improving adversarial robustness from this work.

---

> ### Author Response · Authors · 2022-11-16
> **First reply (part 2)**
>
> ### 3. I am wondering whether this analysis could generalize to modern architectures, e.g. ViT, where BN no longer exists.
>
> Thank you for your valuable suggestions. In the revised paper, we generalize our analysis from vanilla Hybrid-AT with BN to multiple settings where BN no longer exists, such as ViT with Layer Normalization (LN). Additionally, we show the results of ResNet with other variants of normalization, such as Layer Normalization, Group Normalization and Instance Normalization. The experimental results are consistent with the analysis of BN, and corroborate our two-task hypothesis which interprets the main role of dual BN as providing unshared parameters for two tasks. Please refer to Section 5.3 in the revised paper for more details.
>
>
>
>
> | Model  | Norm | Setups    | Branch       | Clean | PGD10 | AA    |
> |--------|------|-----------|--------------|-------|-------|-------|
> | ViT    | LN   | Single AP | AP           | 92.21 | 33.60 | 1.84  |
> |        |      | Dual AP   | AP$_{adv}$   | 58.02 | 30.08 | 12.44 |
> |        |      | Dual AP   | AP$_{clean}$ | 91.60 | 0.00  | 0.00  |
> | ResNet | LN   | Single AP | AP           | 75.12 | 18.81 | 11.80 |
> |        |      | Dual AP   | AP$_{adv}$   | 62.56 | 26.98 | 16.90 |
> |        |      | Dual AP   | AP$_{clean}$ | 88.41 | 0.00  | 0.00  |
> |   | GN   | Single AP | AP           | 81.85 | 21.94 | 14.50 |
> |        |      | Dual AP   | AP$_{adv}$   | 70.27 | 29.36 | 18.30 |
> |        |      | Dual AP   | AP$_{clean}$ | 91.82 | 0.00  | 0.00  |
> |        | IN   | Single AP | AP           | 92.55 | 23.06 | 1.20  |
> |        |      | Dual AP   | AP$_{adv}$   | 52.29 | 25.27 | 16.10 |
> |        |      | Dual AP   | AP$_{clean}$ | 92.35 | 0.00  | 0.00  |
>
> Table 1: Dual AP where BN no longer exists ($\epsilon=16/255$). LN, GN and IN refer to Layer Normalization, Group Normalization and Instance Normalization, respectively.
>
> ### 4. It is recommended to evaluate mixing up NS_adv and NS_clean in Table 1 such that the impact of using more data to estimate BN is eliminated.
>
> Thank you for your valuable suggestions. We follow the recommendation to evaluate mixing up NS$\_{adv}$ and NS$\_{clean}$ and add the results in the revised paper. The results reported in the following Table 2 suggest that the effect of more data to estimate BN by mixing NS$\_{adv}$ and NS$\_{clean}$ (noted as NS$\_{mix}$) is limited. Under the AP$\_{clean}$, the robustness with NS$\_{mix}$ stays zero, which is expected. Under the AP$\_{adv}$, the robustness with NS$\_{mix}$ is slightly lower than the corresponding NS used in training. Note that the gap between NS$\_{adv}$ and NS$\_{clean}$ is not as large as suggested in prior work but still exists, which explains the performance drop of mixed NS during inference (when disentangled NS is applied during training). A similar performance drop caused by this NS inconsistency is also discussed for Trades-AT (See Section 6 in the revised paper).
>
> | Model           | NS           | AP           | Clean | PGD-10 | AA    |
> |-----------------|--------------|--------------|-------|--------|-------|
> | Hybrid-AT       | NS$_{adv}$   | AP$_{adv}$   | 61.84 | 31.67  | 22.51 |
> |                 | NS$_{clean}$ | AP$_{clean}$ | 94.18 | 0.00   | 0.00  |
> |                 | NS$_{mix}$   | AP$_{adv}$     | 61.45 | 29.97  | 18.80 |
> |                 | NS$_{mix}$   | AP$_{clean}$   | 92.00 | 0.00   | 0.00  |
> | Cross Hybrid-AT | NS$_{clean}$ | AP$_{adv}$   | 59.56 | 31.25  | 22.40 |
> |                 | NS$_{adv}$   | AP$_{clean}$ | 93.86 | 0.00   | 0.00  |
> |                 | NS$_{mix}$   | AP$_{adv}$     | 60.09 | 26.41  | 18.36 |
> |                 | NS$_{mix}$   | AP$_{clean}$   | 90.60 | 0.00   | 0.00  |
>
>
> Table 2: Evaluate Hybrid-AT and Cross Hybrid-AT with different NS ($\epsilon=16/255$).

---

> > ### Comment · Reviewer_8GPb · 2022-12-11
> > **Final Recommendation**
> >
> > Dear Authors,
> >
> > Thanks very much for providing additional experiments and explanations. This work revealed some interesting observations in why two batchnorm statistics is effective, i.e. due to the two affine parameters. However, the final recommended practice is to maintain a single batchnorm parameters. There is still a gap between the empirical analysis and the final take home message and the recommended practice is nothing new. Therefore, I would like to keep my original rating.
> >
> > Best
> > Reviewer 8Gpb

---

> > > ### Author Response · Authors · 2022-12-11
> > > **Thanks for your reply**
> > >
> > > Dear reviewer 8Gpb,
> > >
> > > Thank you for your reply.
> > >
> > > The final recommended practice is not to adopt single BN parameters. For example, in the case of vanilla Hybrid-AT, two sets of BN parameters are still necessary. From the two-task perspective, two sets of BN parameters might still be necessary for other AT frameworks or domain adaptation tasks.
> > >
> > > Instead, our final recommended practice is to **not** disentangle the normalization statistics. All our results and analysis refute the prior claims that they should be disentangled as well as the underlying large adversarial-clean distribution mismatch. We believe that our findings and insight are valuable for the community to **understand** and **utilize** dual BN.
> > >
> > > It is appreciated that you spend your time providing feedback on our work.
> > >
> > > Best regards,
> > >
> > > All authors of this work

---

> ### Author Response · Authors · 2022-11-16
> **First reply (part 1)**
>
> Many thanks for the great comments! Please find our replies below.
>
> ### 1. Despite some insights into the effectiveness of affine parameters, there is no novel algorithm proposed.
>
> Thank you for your valuable comments. We agree that there is no new algorithm proposed in this paper. As also recognized by the reviewer, however, our analysis provides insights which can be valuable for future research in the community. In the revised paper, we make every effort to clarify the contributions, especially practical takeaways in Section 5.3 and Section 6. Here, we summarize the modified parts as follows.
>
> **Further clarification of the Two-task hypothesis.** In the originally submitted paper, we propose a two-task hypothesis after pointing out that the clean-adversarial domain gap is not as large as claimed in [1]. In the revised paper, we further discuss the two-task hypothesis by proposing a Dual-Linear model and extending the discussions from BN to more normalization(e.g., Layernorm) and models (e.g. vision transformers). **For the future algorithm design of Hybrid-AT, our two-task hypothesis suggests new possibilities from the multi-task perspective instead of focusing on the misleading domain gap in [1].** Please refer to Section 5.3 for more details.
>
> **Recommended practice of normalization statistics (NS).** In the originally submitted paper, we mainly discussed NS under the setting of dual affine parameters (AP). In the revised paper, we further extend the discussions from dual AP to single AP, and **provide recommended practices of how to design NS in either single or dual AP setting.** Specifically, our improvement of Trades-AT[2] and vanilla Hybrid-AT can be seen as successful practices of improving AT methods with hybrid samples. These practices might not constitute as a novel algorithm, but we believe that they are important messages to the community for the future work on AT with hybrid samples. Please refer to Section 6 for more details.
>
>
>
> [1] Cihang Xie and Alan Yuille. Intriguing properties of adversarial training at scale. ICLR, 2020
> [2] Hongyang Zhang, Yaodong Yu, Jiantao Jiao, Eric P Xing, Laurent El Ghaoui, and Michael I Jordan. Theoretically principled trade-off between robustness and accuracy. In ICML, 2019b.
>
>
>
> ### 2. From practical point of view, is there any recommended practice that one could adopt based on the analysis?
>
> Thank you for your valuable suggestions. In the revised paper, we provide clear recommendations from the practical point of view, which researchers can adopt directly in future work. For the AT methods with hybrid samples, we find that disentangling NS has little influence on the performance when two sets of AP are used. In the case of a single AP, disentangling NS actually harms the performance. As a takeaway, we recommend the practice of NOT disentangling NS for simplicity. Moreover, we also provide possible practices (e.g., KL regularization) to improve the single AP setting, which achieves comparable performance with Dual AP. For more details, refer to the last paragraph of Section 6 in the revised paper.

---

### Official Review · Reviewer_uM2g · 2022-10-25

**Confidence:** 4
**Clarity, Quality, Novelty And Reproducibility:** Please see my comments above.
**Correctness:** 2
**Technical Novelty And Significance:** 2
**Empirical Novelty And Significance:** 2
**Recommendation:** 5

**Strength And Weaknesses:**

Strength:
This paper reconsiders the motivation of applying dual BN in adversarial training with hybrid samples and explores the effectiveness of normalization statistics and affine parameters separately.

This paper provides a closer looker at dual BN: disentangling the mixture distribution for normalization statistics and introducing two sets of affine parameters (AP). Then they find that the performance improvement mainly comes from more affine parameters.

The authors point out a visualization flaw in previous work: when AP is fixed, the robustness gap between the NS calculated on clean or adversarial samples is quite subtle.


Weakness:

This paper lacks a description of the existing development and problems in the field of applying batch normalization in adversarial training. It is not clear why they want to explore the reasonableness of the motivation for applying dual batch normalization proposed in the previous work. For example, is there some bad phenomenon with the application of dual batch normalization? or some limitations not considered in previous work? In a word, the authors do not give a reason why they carry out this work.

The cross BN model looks like a mixture of ordinary BN and Dual BN. There is not much to take in this model. Although the authors provided two variants: setup1 and setup2, the contribution still looks incremental.

In Figure 2, Cross-AT method has a similar performance to Hybrid AT. However, from the statement that “the cross-AT still keeps its clean branch in the Hybrid-AT.., but the model weights are updated only by the adversarial branch’’, Cross-AT does not use clean samples to update the model weights, how does its performance on clean samples achieve the similar with Hybrid AT?

In the last paragraph of section 5, the paper claims that “future works utilizing dual BN in Hybrid AT are suggested to discard the two-domain hypothesis and embrace the two-task hypothesis”, the authors emphasize they have two intriguing findings. However, the authors have not clarified what contributions this work will bring to the development of the related field.

In Page 2 Line 13, the authors claim that “This conjecture is motivated by an interesting observation This conjecture is motivated by an interesting observation that swapping NS in the Hybrid-AT has little influence on the model performance”. Although the experiment results shown in Figure 2 may prove the little influence of Normalization statistics, the experiment setting itself is hard to interpret. That is, the authors may not illustrate what motivates them to this experiment or if is there any interpretable meaning about the operation of swapping NS.

In the last paragraph of section 3, the authors claim that “Such a success inspires us to suspect that the merit of dual BN over single BN in Hybrid-AT …”. However, as the results shown in Figure 2, although there is no significant drop in performance after “Cross-AT”, there is also no significant gain to prove the effectiveness of “Cross-AT”. Therefore, the claim of “success” may not be accurate.

In table 2, although the test accuracy of Setup1(AP_{adv}) has improved compared to Single BN and Dual BN in PGD-10 and AA, the test accuracy has obviously dropped in the clean dataset. So the claim on Page 2 Line 2 “two sets of APs alone significantly improve the robustness.” may be weakly supported by the experiment.

The paper conducts a lot of experiments but lacks some necessary theoretical analysis in dual BN and Hybrid-AT. Moreover, a novel effective method is expected according to the theoretical finding.


Minor issues:

In the second contribution listed in section 1: “We point out that it also introduces two sets of AP”, what“it”refers to is not clear.

In section 2.1, the part named “Experimental setups” is expected to be close to the experiment result in section 4 or section 5 of the article.

In Table 2, Table 3, there is no description of symbolic subscripts of AP_adv  AP_clean,.



**Summary Of The Paper:**

This paper studies the motivation of applying dual BN in a hybrid adversarial training scene, where the model is trained on both adversarial samples and clean samples. In contrast to the popular belief that normalization statistics should be estimated separately for clean samples and adversarial samples to achieve stronger robustness, this paper reveals that what makes dual BN effective lies in two sets of affine parameters. Moreover, the paper demonstrates that the adversarial-clean domain gap is not as large as many might expect and it is similar to its noisy-counterpart under the same perturbation/noise magnitude. Overall, this paper raises an interesting topic but the algorithm part is a bit weak.

**Summary Of The Review:**

The contribution of the paper is limited and the reason why exploring the reasonableness of the motivation for applying dual batch normalization proposed in the previous work is not clear.

---

> ### Author Response · Authors · 2022-11-16
> **First reply (part 2)**
>
> ### 6. The claim of “success” in the last paragraph of section 3 may not be accurate
>
>
>
> Thank you for your suggestions. We replace the "success" with "compatibility of BN$\_{clean}$ with the adversarial branch" in the revised paper. We highlight that it is not our intention to achieve superior performance with Cross-AT and follow the suggestion to drop the term "success" to avoid this impression.
>
>
>
> ### 7. On the claim “two sets of APs alone significantly improve the robustness.” on Page 2 Line 2.
>
> Originally, “two sets of APs alone significantly improve the robustness.” was claimed to highlight that, without disentangling NS, two sets of APs improve the robustness over the single BN. In other words, two sets of APs achieve comparable performance as dual BN (which disentangles NS and uses two sets of APs). We agree that this sentence could be confusing and have revised it in the paper.
>
>
>
> ### 8. The paper conducts a lot of experiments but lacks some necessary theoretical analysis in dual BN and Hybrid-AT. Moreover, a novel effective method is expected according to the theoretical finding.
>
>
> Our work is mainly inspired by prior work [1] which is the pioneering work to propose dual BN in Hybrid-AT. After demonstrating the effectiveness of dual BN, prior work [1] proposes a two-domain hypothesis for the theoretical justification of dual BN. In contrast to prior work [1], we propose a two-task hypothesis for theoretical justification. In the revision, we have further verified the two-task hypothesis with dual linear classifier experiment and extended it to other architectures and normalization modules without BN. For more details, see section 5.3 in the revised paper.
>
> For the future algorithm design of Hybrid-AT, our two-task hypothesis suggests new possibilities from the multi-task perspective instead of focusing on misleading domain gaps in [1]. The theoretical insight with the two-task hypothesis partly inspires our further investigation at Section 6 in the revised paper. As a result, we find that simple practices might be promising for improving Hybrid-AT with a single BN. These practices might not constitute as a novel algorithm, but we believe that both the two-task hypothesis and recommended practices are important and insightful messages to the community for the future work on AT with hybrid samples. Please refer to Section 6 for more details.
>
> [1] Cihang Xie and Alan Yuille. Intriguing properties of adversarial training at scale. ICLR, 2020
> [2] Hongyang Zhang, Yaodong Yu, Jiantao Jiao, Eric P Xing, Laurent El Ghaoui, and Michael I Jordan. Theoretically principled trade-off between robustness and accuracy. In ICML, 2019b.
>
>
>
> ### 9. Minor issues:
>
> #### 9.1. In the second contribution listed in section 1: “We point out that it also introduces two sets of AP”, what“it”refers to is not clear.
>
> Thank you for your valuable suggestions. "It" refers to "dual BN". However, we have resummarized the contribution part in the revised paper.
>
>
> #### 9.2. In section 2.1, the part named "Experimental setups" is expected to be close to the experiment result in section 4 or section 5 of the article.
>
> Thank you for your valuable suggestions. For a coherent storyline, we have summarized the original "Experimental setups" in the introduction and moved the details to the appendix in the revised paper.
>
> #### 9.3. In Table 2, Table 3, there is no description of symbolic subscripts of AP$\_{adv}$ and AP$\_{clean}$.
> Thank you for your valuable  suggestions. The subscripts of AP$\_{adv}$ and AP$\_{clean}$ indicate the input data type used during training. We have added the descriptions of AP$\_{adv}$ and AP$\_{clean}$ in the Table title in the revised paper.

---

> ### Author Response · Authors · 2022-11-16
> **First reply (part 1)**
>
> Many thanks for the great comments! Please find our replies below.
>
> ### 1. In a word, the authors do not give a reason why they carry out this work (is there some bad phenomenon with the application of dual batch normalization? or some limitations not considered in previous work? )
>
> A drawback of applying dual BN in Hybrid lies in the unknown source of samples, which makes it difficult to choose BN during inference. Prioritizing the robustness, prior work [1] adopts BN$\_{adv}$ at test time at the cost of clean accuracy drop. Moreover, Prior work interprets the necessity of dual BN from the perspective of an inherent large adversarial-clean domain gap, which implicitly suggests disentangling NS (via dual BN) might be the only solution. Our work revisits how dual BN works in Hybrid-AT and finally suggests a new two-task hypothesis, which motivates an alternative single BN solution from the regularization perspective. We discuss the drawbacks of prior work in the last part of related work. Moreover, the drawback is discussed in Section 6 of the revised paper, where we show that, with a careful choice of training details, a single BN can achieve competitive performance and therefore avoid the choice of BN during inference.
>
>
>
>
>
>
> ### 2. Discussions on Cross BN model and Setup1/2
>
> We clarify that neither Cross BN nor Setup1/Seup2 is not a new method. The intention of designing these experiments is to demonstrate what component in dual BN makes it suitable for Hybrid-AT. Specifically, Cross BN experiment setup motivates our conjecture that what makes dual BN effective lies in two sets of AP instead of disentangled NS. This conjecture is verified by untwining NS and AP in Setup1 and Setup 2.
>
>
>
>
> ### 3. How Cross-AT achieves similar performance with Hybrid-AT without using clean samples to update the model weights?
>
>
> Thank you for your valuable comments. We clarify that there is a BN choice during inference for Hybrid-AT with dual BN: when BN$\_{clean}$ is adopted during inference, the model has high accuracy on clean samples but zero robustness; when BN$\_{adv}$ is adopted during inference, the model has high robustness but low accuracy on clean samples (See Table 1 of the revised paper for the corresponding results). Since we are mainly interested in whether BN$\_{clean}$ can be compatible with the adversarial branch to achieve robustness, in Figure 2, we compare Cross-AT to the performance Hybrid-AT with BN$\_{adv}$ being adopted during inference. Note that the high accuracy of Hybrid-AT is only available when BN$\_{clean}$ is adopted during inference. This well explains why Cross-AT with the model weights being updated only by the adversarial branch achieves a similar accuracy as that of Hybrid AT (BN$\_{adv}$ at inference). We have added more clear descriptions of Figure 2 in the revised paper.
>
>
>
> ### 4. Discussions on what contributions this work will bring to the development of the related field
>
>
> Thank you for your valuable comments. We have revised the last paragraph of Section 5 and extended it to a new subsection for verifying the two-task hypothesis which further allows more investigation on other architectures without BN. Moreover, we have added Section 6 for additionally investigating Trades-AT and clarifying how our findings can be used as a takeaway which is valuable to the development of the related field.
>
> ### 5. What motivates us to the Cross-AT experiment or is there any interpretable meaning about the operation of swapping NS?
>
>
> Thank you for your valuable comments. What motivates the Cross-AT experiment is to seek an answer to the following question: Can BN$\_{clean}$ be compatible with the adversarial branch to yield a robust model during inference? We are interested in answering the above question because it has significant implications on whether domain-specific BN statistics are better than the mixed ones for Hybrid-AT. An interpretable meaning of swapping the BN statistics from adversarial to clean is as follows: it constitutes using full cross-domain BN statistics. If the merit of dual BN over single BN lies in replacing mixed BN statistics with domain-specific ones, i.e., totally avoiding cross-domain statistics, Cross-AT with full cross-domain BN statistics might be expected to yield very low robustness. We have revised Section 3 to illustrate the above motivation and interpretable meaning of our preliminary Cross-AT investigation.

---

### Official Review · Reviewer_UFTM · 2022-10-25

**Confidence:** 3
**Clarity, Quality, Novelty And Reproducibility:** see the strength and weakness.
**Correctness:** 3
**Technical Novelty And Significance:** 2
**Empirical Novelty And Significance:** 3
**Recommendation:** 5

**Strength And Weaknesses:**

\+ Through untwining NS and AP in dual BN, it demonstrates that what makes it effective lies in two sets of AP instead of disentangled NS (as claimed in prior work).

\+ It points out a hidden flaw in prior work for visualizing the NS to highlight a (large) domain gap between adversarial and clean samples. After fixing it, it reveals that the adversarial-clean domain gap is not as large as prior work suggests.

\- The technical contribution may be limited. This paper perform a lot of experiments with the dual BN for Hybrid-AT. It tests many different AT and BN settings. But it mainly uses Hybrid-AT techniques and variants of BN. The technical contribution may be limited.

\- The application of the proposed method may be limited. The model must have BN layers so that it can be applied. Besides, it basically have two paths in the model for adversarial and clean data. During training, it knows how to activate each path since the model knows whether the data is clean or adversarial. But during inference without knowing the data is clean or adversarial, how can the model know which path to activate?

**Summary Of The Paper:**

This paper firstly attempts at training a model with cross-domain BN statistics. Through untwining NS and AP in dual BN, it demonstrates that what makes it effective lies in two sets of AP instead of disentangled NS (as claimed in prior work).
It points out a hidden flaw in prior work for visualizing the NS to highlight a (large) domain gap between adversarial and clean samples. After fixing it, it reveals that the adversarial-clean domain gap is not as large as prior work suggests.

**Summary Of The Review:**

It mainly compares with the work (Xie & Yuille, 2020) and points out some key insights different from Xie & Yuille, 2020. It performs comprehensive experiments to check certain assumptions. But it mainly use existing techniques. The technical contribution may be limited. And the application may be limited. It may still have problems to apply this method in practice.

---

> ### Author Response · Authors · 2022-11-16
> **First reply**
>
> Many thanks for the great comments! Please find our replies below.
>
>
> ### 1. More comprehensive discussions beyond vanilla Hybrid-AT with BN
>
> We understand the concerns of Reviewer UFTM. In the revised paper, we generalize our discussions in vanilla Hybrid-AT with Dual BN to more methods (e.g., Trades-AT),  normalization (e.g., Layer Normalization), and models (e.g., vision transformer) for a comprehensive analysis. These results are consistent with the findings in vanilla Hybrid-AT with Dual BN. We summarize the main results and findings as follows. Please refer to Section 5.3 and Section 6 in the revised paper for more details.
>
> #### 1.1. Results beyond vanilla Hybrid-AT
> Beyond Hybrid-AT, we discuss Trades-AT which is another variant of AT with hybrid samples. We point out that the official implementation of Trades-AT with a single BN implicitly disentangles NS during training but uses mixed NS during inference, which results in unintended NS inconsistency. After fixing this unintended inconsistency by sending clean and adversarial samples together to the model, we improve the performance by a large margin, as shown in the following Table 1, where Algorithm 1 and Algorithm 2 are detailed in Appendix D. This further motivates a KL regularization loss to improve the performance of vanilla hybrid-AT with single BN. For more details, see section 6 in the revised paper.
>
> | Trades-AT                | Clean | PGD10 | AA    |
> |--------------------------|-------|-------|-------|
> | Single BN (Algorithm 1, original)              | 69.97 | 26.8  | 18.04 |
> | Single BN (Algorithm 2, ours)   | 75.14 | 29.41 | 21.32 |
> | Dual BN (BN$_{adv}$)     | 62.15 | 33.16 | 25.67 |
> | Dual BN (BN$_{clean}$)   | 78.03 | 16.17 | 2.21  |
>
> Table 1: A closer look at Trades-AT with different BN settings, $\epsilon=16/255$.
>
>
> #### 1.2. Results where BN does not exist
> In order to generalize our findings to more diverse settings where BN does not exist, such as ResNet with Layer Normalization (LN). We also experiment with recent ViT that uses LN  by default. We report the results of different normalization and models in the following Table 2. The experimental results are consistent with the analysis of BN, and corroborate our two-task hypothesis which interprets the main role of dual BN as providing unshared parameters for two tasks. Please refer to Section 5.3 in the revised paper for more details.
>
>
> | Model  | Norm | Setups    | Branch       | Clean | PGD10 | AA    |
> |--------|------|-----------|--------------|-------|-------|-------|
> | ResNet | LN   | Single AP | AP           | 75.12 | 18.81 | 11.80 |
> |        |      | Dual AP   | AP$_{adv}$   | 62.56 | 26.98 | 16.90 |
> |        |      | Dual AP   | AP$_{clean}$ | 88.41 | 0.00  | 0.00  |
> |   | GN   | Single AP | AP           | 81.85 | 21.94 | 14.50 |
> |        |      | Dual AP   | AP$_{adv}$   | 70.27 | 29.36 | 18.30 |
> |        |      | Dual AP   | AP$_{clean}$ | 91.82 | 0.00  | 0.00  |
> |        | IN   | Single AP | AP           | 92.55 | 23.06 | 1.20  |
> |        |      | Dual AP   | AP$_{adv}$   | 52.29 | 25.27 | 16.10 |
> |        |      | Dual AP   | AP$_{clean}$ | 92.35 | 0.00  | 0.00  |
> | ViT    | LN   | Single AP | AP           | 92.21 | 33.60 | 1.84  |
> |        |      | Dual AP   | AP$_{adv}$   | 58.02 | 30.08 | 12.44 |
> |        |      | Dual AP   | AP$_{clean}$ | 91.60 | 0.00  | 0.00  |
>
> Table 2: Dual AP where BN no longer exists ($\epsilon=16/255$). LN, GN and IN refer to Layer Normalization, Group Normalization and Instance Normalization, respectively.
>
>
>
>
> ### 2. More discussions on the drawback of dual BN and how to solve it for practical application.
>
> We agree that the BN path cannot be chosen according to the data type during inference since the data type is unknown. However, we point out that this is the inherent drawback of Dual BN [1], and the authors adopt BN$_{adv}$ for high robustness during inference at the cost of clean accuracy drop in official Dual BN[1]. This drawback is summarized in the final part of the related work. Moreover, we revisit another variant of AT with Hybrid samples (Trades-AT[2]). We show that, with a careful choice of training details, a single BN might be sufficient for achieving competitive performance. Our discussions provide recommended practices on how to handle the norm layers in AT with hybrid samples. For more details, see section 6 in the revised paper.
>
> ### 3. On contribution regarding existing techniques.
> This work focuses on providing an extensive analysis of existing techniques. Such analysis and investigation yield small modifications to existing techniques but cause a non-trivial performance boost (see Table 8 and Table 9 in the revised paper).
>
>
> [1] Cihang Xie and Alan Yuille. Intriguing properties of adversarial training at scale. ICLR, 2020.
>
> [2] Hongyang Zhang, Yaodong Yu, Jiantao Jiao, Eric P Xing, Laurent El Ghaoui, and Michael I Jordan. Theoretically principled trade-off between robustness and accuracy. In ICML, 2019b.

---

### Official Review · Reviewer_dNrE · 2022-11-04

**Confidence:** 3
**Clarity, Quality, Novelty And Reproducibility:** please see above
**Correctness:** 4
**Technical Novelty And Significance:** 3
**Empirical Novelty And Significance:** Not applicable
**Recommendation:** 6

**Strength And Weaknesses:**

Overall, I think the message conveyed by this paper is simple yet important: The disentangled affine transformation is what really matters in the Mixture BN, instead of the disentangled BN statistics as previously believed by the community. The experimental results effectively support the conclusions. The paper is also well-written and is easy to follow for readers without expert knowledge in Mixture BN.

I have one curious question for the authors:

In order to provide more evidence that clean and adversarial samples have similar BN statistics when we don't use disentangled affine transformation, could you please show similar visualization results as those in Figure 4, but remove all affine transformations in all BN layers? This can be achieved by disabling affine transformation when defining PyTorch BN layers. Usually, in normal training, disabling the affine transformations in BN layers won't harm performance much.


**Summary Of The Paper:**

This paper makes the interesting observation that the disentangled affine transformation is what really matters in the Mixture BN, instead of the disentangled BN statistics as previously believed by the community.

**Summary Of The Review:**

please see above

---

> ### Author Response · Authors · 2022-11-16
> **First reply**
>
> Many thanks for the great comments! Please find our replies below.
>
> ### Visualization results of statistics when affine parameters are disabled during training.
>
> Thank you for your valuable suggestions. Following your suggestion, we visualize normalization statistics when all affine parameters are disabled during training (and inference) and add the result in the revised paper (please refer to Figure 8 in the appendix). As shown in Figure 8,  the normalization statistics of clean and adversarial samples are quite similar. With AP disenabled, it is equivalent to the scenario that the two branches have the same set of AP (with $\gamma$ being one and $\beta$ being zero). Therefore, the results when AP is disenabled collaborate with the finding that, with the same set AP, adversarial-clean domain gap is not as large as suggested in prior work.
>
> During this rebuttal, we have further improved the paper's quality, such as extending the two-task hypothesis to more architectures where BN no longer exists (e.g., vision transformer with Layer Normalization). We have additionally added a new section by revisiting Trades-AT (another variant of AT with hybrid samples) with takeaway insight. Please check the revision summary and our revised paper.

---

### Author Response · Authors · 2022-11-16
**Paper revision summary**

We thank all reviewers for the insightful feedback. Overall, it seems that all reviewers recognize the strengths of our work in pointing out that affine parameters play the main role in Dual BN of Hybrid-AT and the clean-adversarial domain gap is not as large as claimed in prior work. According to all the valuable suggestions, we make every effort to improve our paper during the rebuttal, and summarize the main revisions as follows. Please refer to our revised paper for more details. The modified content is indicated with red color while the newly added content is indicated with blue color.

### 1. On the practical implications of this paper

#### 1.1. Extended results with the Two-task hypothesis
In the revised paper, we have further verified the two-task hypothesis with a dual linear classifier experiment and extended it to other architectures (e.g., vision transformer) and normalization modules (e.g., Layer Normalization) where BN does not exist. More normalization results are reported in the appendix. The two-task perspective and all the experimental results suggest new possibilities for future algorithm design from the multi-task perspective instead of focusing on the misleading domain gap as claimed in prior work. For more details, see section 5.3 in the revised paper.

#### 1.2. Extended investigation of Hybrid-AT beyond its vanilla version and takeaway insights
The two-task hypothesis inspires us for further investigation beyond vanilla Hybrid-AT in the revised paper. We extend the analysis under the Dual BN setting to Trades-AT (another variant of Hybrid-AT) which has a single BN by default. As takeaways, we provide recommended practices of how to design NS in either single or dual AP setting. We believe the recommended practices are important messages to the community for future work on AT with hybrid samples. Please refer to Section 6 for more details.

### 2. On the drawback of dual BN and interpretable meaning of NS swapping
#### 2.1. Describing the problem of applying dual BN in practice
A drawback of applying dual BN in Hybrid-AT lies in the unknown source of samples, which makes it difficult to choose BN during inference. Prior work interprets the necessity of dual BN from the perspective of an inherent large adversarial-clean domain gap, which implicitly suggests disentangling NS (via dual BN) might be the only solution. Our work revisits how dual BN works in Hybrid-AT and finally suggests a new two-task hypothesis, which motivates an alternative single BN solution with appropriate regularization.
#### 2.2. Clarifying the interpretable meaning of NS swapping
An interpretable meaning of swapping the BN statistics from adversarial to clean is as follows: it constitutes using full cross-domain BN statistics. If the merit of dual BN over single BN lies in replacing mixed BN statistics with domain-specific one, i.e., totally avoiding cross-domain statistics, Cross-AT with full cross-domain BN statistics might be expected to yield very low robustness.

The above content has been added in the revised paper and is indicated with blue color.

### 3. Modifications of introduction and moving some content into appendix.
Due to the added new content, we have restructued the introduction, resummarized the contributions and moved some parts of the content in the original submission to the appendix for saving space. A small modicfication is also applied to the abstract and conclusion.


**Overall, we express our genuine gratitude to all the reviewers for helping enhance this work to a new level.**

**If anything remains unclear to the reviewers, please kindly let us know.**

---

### Author Response · Authors · 2022-11-18
**Paper revision summary (second-round)**

We thank all reviewers for the insightful feedback again.

The main change in the second revision is to include Appendix D to discuss the implementation of Trades_AT with the pseudo-code provided.

**Could we request the reviewers to take the time to check our rebuttal as well as our revised paper to confirm whether anything remains unclear? If there are any other concerns, please kindly let us know.**

---

### Author Response · Authors · 2022-11-19
**Thanks to all reviewers and ACs**

During this rebuttal period, we have spent every effort to improve the paper's quality and address the concerns of the reviewers. As the paper revision deadline is coming soon, we might not be able to further revise the paper but we will continue to reply to the reviewers/ACs to address new concerns if any.

**It is highly appreciated that the reviewers can take the time to check our rebuttal as well as our revised paper.**

**We thank all reviewers and ACs for their effort on evaluating this work.**

---

### Decision · Program_Chairs · 2023-01-20

**Decision:**

Reject

**Justification For Why Not Higher Score:**

Limited impact.

**Justification For Why Not Lower Score:**

N/A

**Metareview: Summary, Strengths And Weaknesses:**

We have set up a video call with reviewers to discuss this work and all reviewers agree that the observation and analysis is interesting and inspiring. However, it is hard to directly use this understanding to improve robustness in practice, which greatly limits the power and contribution of this work to a broader research community.

**Summary Of Ac-Reviewer Meeting:**

The major concern of reviewers is the developed understanding of batch normalization for clean and adversarial examples can be hardly generalized to benefit adversarial robustness in practice. Showing more real cases to improve robustness based on this understanding can greatly improve the contributions of this work.